# Differences between Obese and Non-Obese Children and Adolescents Regarding Their Oral Status and Blood Markers of Kidney Diseases

**DOI:** 10.3390/jcm10163723

**Published:** 2021-08-21

**Authors:** Katarzyna Maćkowiak-Lewandowicz, Danuta Ostalska-Nowicka, Jacek Zachwieja, Elżbieta Paszyńska

**Affiliations:** 1Department of Pediatric Nephrology and Hypertension, Poznan University of Medical Sciences, 60-572 Poznan, Poland; dostalska@ump.edu.pl (D.O.-N.); jacekzachwieja@ump.edu.pl (J.Z.); 2Department of Integrated Dentistry, Poznan University of Medical Sciences, 60-812 Poznan, Poland; paszynska@ump.edu.pl

**Keywords:** obesity, childhood, dental caries, gingivitis, kidney injury, glomerulopathy, uric acid, cystatin C

## Abstract

(1) Background: A rarely discussed effect of obesity-related glomerulopathy (ORG) may slowly lead to irreversible glomerular damage and the development of chronic kidney disease. These patients need to undertake medical care, but whether they should be included in intensive oral care is still not mandatory. The study aimed to assess a relationship between renal, metabolic, and oral health indicators among pediatric patients affected by simple obesity. (2) Methods: 45 children and adolescents with simple obesity hospitalized (BMI 34.1 ± 4.8 kg/m^2^, age 15.4 ± 2.3) and compared with 41 aged-matched healthy controls (BMI 16.4 ± 2.4 kg/m^2^, age 15.4 ± 2.7). Echocardiography, 24-h ambulatory blood pressure monitoring, ultrasound exam with Doppler, and laboratory tests including kidney and metabolic markers were performed. Oral status was examined regarding the occurrence of carious lesions using decay missing filling teeth (DMFT), gingivitis as bleeding on probing (BOP), and bacterial colonization as plaque control record (PCR). (3) Results: The strongest correlation was revealed between BMI and concentration of uric acid, cystatin C, GFR estimated by the Filler formula (r = 0.74; r = 0.48; r = −0.52), and between oral variables such as PCR and BOP (r = 0.54; r = 0.58). Children and adolescents with obesity demonstrated untreated dental caries, less efficient in plaque control and gingivitis. (4) Conclusions: No specific relation to markers of kidney disease were found; however, more frequent gingivitis/bacterial colonization and significant differences in oral status between obese and non-obese patients were revealed. Susceptibility to inflammation may be conducive to developing metabolic syndrome and kidney damage in the form of obesity-related glomerulopathy and contribute to future dental caries. Uric acid seems to indicate metabolic syndrome and cardiovascular complications (LVMI > 95 percentiles). Cystatin C and uric acid might aspire to be early markers of kidney damage leading to obesity-related glomerulopathy.

## 1. Introduction

The COVID-19 outbreak, which contributed to a decrease in physical activity and strengthening of harmful eating habits resulting from remote learning and lockdown, has significantly accelerated an increase in the number of obese patients in Poland. It seems to be a necessary task of the medical environment to address a high risk of obesity complications among pediatric patients with obesity. Typical complications of obesity in children and adolescents, for example, arterial hypertension and dyslipidemia, are well known. A rarely discussed effect is obesity-related glomerulopathy (ORG). Despite its initial asymptomatic course, it slowly leads to irreversible glomerular damage and development of chronic kidney disease, with end-stage renal failure in 1 per 3 patients. In the first period of ORG, obesity leads to hyperfiltration. An increased tubular flow decreases the contact time between proteins and tubular epithelium and produces an increased radial gradient of albumin concentration, resulting in reduced protein reabsorption that may occur without evidence for glomerular dysfunction. The excessive reabsorption of proteins generates apoptosis, renal oxidative stress, inflammation, hypoxia, and lipid accumulation, leading to fibrosis [1,2]. Clinically, ORG is manifested with proteinuria of various severity, depending on the extension of glomerular structural changes: from permanent, mild proteinuria to nephrotic proteinuria [3]. A final diagnosis of obesity-related glomerulopathy is based on biopsy, which carries a higher risk of complications than standard diagnosis due to the patient’s obesity and more difficult access to the material collected during a biopsy [4]. It is not a routine procedure and selecting a group of ORG patients without this type of invasive diagnostic procedure is a big challenge for physicians [5]. ORG is suspected in obese patients with proteinuria and concomitant arterial hypertension and lipid metabolism disorders [6,7]. Significant underestimation of ORG patients stems from the fact that markers of this disease revealed by thin-needle biopsy are also found in patients without proteinuria [8]; therefore, new, early markers of kidney damage that would find use in diagnosing ORG are searched for, such as urine megalin and expression of connexin 43 [9,10]. Pediatric patients affected by obesity should be encouraged to undertake medical care, but whether they should be included in oral care because of dental or gingival diseases remains still an uncertain requirement; however, more and more information concerning overlapping relationships with oral cavity homeostasis is available in the literature. Proinflammatory cytokines, produced by adipose cells, lead to low-grade chronic inflammation in the salivary glands [11]. The cause of the imbalance of salivary substances is a significant increase in susceptibility to caries, even in very young patients [12]. A diet rich in carbohydrates favors bacterial colonization, creating favorable conditions for caries promotion. Another oral manifestation of obesity may take the form of gingival inflammation due to poor oral hygiene. Then gingivitis may generate periodontal tissues response and intensify metabolic disorders or even promote weight gain. Combining the above-mentioned dental complications of obesity, such as dental caries, microbial imbalance, and defective salivary composition stimulates potential risk for systemic diseases [11]. It will most likely be connected with an inflammatory response to bacterial pathogens and the generation of reactive oxygen species. Oxidative imbalance contributes to generating lipid peroxide, which can diffuse into the bloodstream [12]. The components of the oxidative stress cascade interact with glomeruli and contribute to obesity-related glomerulopathy [12].

The aim of the study was to assess a correlation between renal, metabolic, and oral health indicators among pediatric patients affected by simple obesity.

## 2. Materials and Methods

### 2.1. Study and Control Groups

The study group finally consisted of 45 children and adolescents with simple obesity, admitted to the Department of Pediatric Nephrology and Hypertension in Poznań between 1 January 2019 and 31 December 2020. A reason for hospitalization was the verification of primary hypertension. The control group consisted of 41 age-matched healthy children and adolescents. The flow chart of the study is presented as Appendix A.

Echocardiography was performed on all patients with the assessment of left ventricular hypertrophy marker (left ventricular mass index—LVMI > 95 centile), as were abdominal ultrasound exam with Doppler test to evaluate the blood flow through kidneys’ arteries and blood laboratory analysis to exclude comorbidities [13]:Peripheral blood morphology, blood gases, C-reactive protein.Concentrations of creatinine, cystatin C, uric acid (UA), urea, Na, K, Ca, Mg.Total cholesterol, LDL-cholesterol, HDL-cholesterol, triglycerides, glucose, glycated hemoglobin, insulin profile.Thyroid-stimulating hormone, triiodothyronine.Liver enzymes, brain natriuretic peptide, troponin.Urine analysis, 24-h urine concentrations of protein and albumin, Na.

All patients were fasting for twelve hours for an ultrasound exam and laboratory tests. During next twenty four hours, urine collection was carried out by collecting every urine sample in a container, starting with the second urinating on the first day of hospitalization, ending the next day with the first portion of urine. Laboratory tests were performed using a standardized method, according to the manufacturer’s instructions. Glomerular filtration rate was estimated by Schwartz and Filler formula [14]. Based on the Filler formula, hyperfiltration was defined as GFR > 130 mL/min/1.73 m^2^ and hypofiltration when GFR was less than 90 mL/min/1.73 m^2^. Arterial blood pressure was assessed by 24-h ambulatory blood pressure monitoring [15]. Atrial hypertension was diagnosed when average systolic or diastolic blood pressure or both, obtained in the course of over three different medical visits was at/above the 95th percentile for age, sex, and height [16]. Simple obesity (BMI > 95 percentiles, BMI Z-score  >  1.4) was diagnosed when secondary obesity was excluded. BMI values were transformed into BMI Z-scores using WHO reference values for pediatric BMI [17].

Inclusion and exclusion criteria for study and control groups are described in Table 1 [18]. The protocol allowed 45 children and adolescents with simple obesity; 25 patients additionally with primary hypertension within this group; therefore, the obesity group was divided into two subgroups: suspicion of obesity-related glomerulopathy (sORG) and without ORG group (without ORG). The selection of obesity patients with sORG was based on scientific reports [4,7]: the presence of significant clinical microalbuminuria (>30 mg/day), primary hypertension (revealed during hospitalization), and lipid disorders (decreased concentration of HDL and increased concentration of triglycerides). Seven adolescents met the whole inclusion criteria for suspicion of obesity-related glomerulopathy group. Forty-one healthy children and adolescents enrolled in the control group had to fulfill the inclusion criteria presented in Table 1.

### 2.2. Dental Examination

A dental assessment was carried out at the dental office by one qualified dentist (E.P.) during hospitalization in the Department of Pediatric Nephrology and Hypertension, in a blind fashion for each child, before the final diagnosis. Before this study, the dental examiner was calibrated after a training course evaluation described in a previous project [19]. Dental evaluation of the occlusal, buccal, and lingual teeth surfaces was performed after the cleaning and drying (excluding the third molars) under visual and tactile examination in artificial light, without magnification, using a dental mirror and blunt probe [20]. The dental assessment included the number of milk/permanent (using lower case letter for primary teeth/capital letters for permanent teeth) decayed teeth (d/D), the number of missed teeth (m/M), the number of filled teeth (f/F), as a dmft/DMFT score [20]. Active dental caries was scored when the lesion manifested a visible cavity, undermined enamel, or softened area. A tooth rebuilt due to dental caries was recorded when a tooth had at least one final restoration applied to cure decay. The missing constituent of the DMFT index was estimated when a tooth had been removed due to dental caries failures. A manual graded periodontal explorer assessed dental plaque and gingival inflammation (LM-instruments, LM8 5050 probe, Osakeyhtiö, Parainen, Finland). Plaque deposit was estimated by the plaque control record index (PCR) [21]. Gingival inflammation was determined using the bleeding on probing index (BOP) [22], measured in six points of the gingival sulcus of all teeth (excluding the third molars). The proportion of surfaces (%) with dental plaque or bleeding-on-probing gums, respectively, were calculated for each subject as % of sites [23]. STROBE protocol was included and attached in the Appendix A.

### 2.3. Statistical Analysis

The Shapiro–Wilk test was used to assess the normality of the data. The homogeneity of variance of each variable was calculated with Levene’s test. Non-parametric Mann–Whitney U-test was applied in the analyses of data with non-normal distribution. Student t-test was employed in the analyses of variables with normal distribution. Spearman’s correlation rank test was performed to analyze the correlation of parameters with non-normal distribution. Pearson test was employed to test the correlation of normal distribution variables. The statistical significance level was set at *p* < 0.05. Statistical analyses were conducted using Statistica 13.3 (TIBCO Software Inc., Palo Alto, CA, USA).

## 3. Results

This section may be divided into subheadings. It should provide a concise and precise description of the experimental results, their interpretation, and the empirical conclusions that can be drawn.

### 3.1. Anthropometric, Clinical, and Biochemical Evaluation

The final group consisted of 86 participants (45 with obesity and 41 control subjects). The mean age of the obese patients was 15.4 ± 2.33 years, the mean age of the controls was 15.4 ± 2.74 years, with no statistically significant difference. BMI and BMI Z-score were statistically different between patients and controls (*p* = 0.001; *p* = 0.001). The clinical and biochemical characteristics of all patients are presented in Table 2.

The study and control group did not differ in serum concentrations of creatinine and urea. Significantly higher concentrations of uric acid and cystatin C were observed in obesity group (*p* = 0.001; *p* = 0.028). According to the Schwartz formula, GFR values did not differ significantly between the study and control group (115.57 ± 20.54 mL/min/1.73 m^2^ and 118.02 ± 36.02 mL/min/1.73 m^2^, respectively). Mean GFR estimated by Filler formula in the obesity group was 111.26 ± 31.15 mL/min/1.73 m^2^ and 127.18 ± 22.54 mL/min/1.73 m^2^ in the control group (*p* = 0.019). The study and control group presented no proteinuria in urine analysis. Clinically significant microalbuminuria (>30 mg/d) was observed in 7 patients with obesity, while in the control group, microalbuminuria >30 mg/d was not detected (*p* = 0.183).

In the study group, 25 patients presented primary hypertension, while AH was not observed in the control group. Obesity and control groups did not differ in serum concentrations of total cholesterol and LDL. Significantly higher concentrations of HDL, triglycerides and glucose were observed in study group (*p* = 0.001; *p* = 0.001; *p* = 0.029). Median LVMI was significantly different between obesity and control groups (*p* = 0.001).

### 3.2. Oral Data

Analysis of the oral cavity revealed that obesity group presented a significantly higher number of D (*p* = 0.001) and DMFT score (*p* = 0.016), amount of dental plaque (PCR), (*p* = 0.001), and gingival inflammation (BOP), (*p* = 0.001). No differences in the results of the dental examination of the primary dentition were detected between the study and the control group (Table 2).

### 3.3. Correlations between the Variables

In the obesity group, Spearman analysis evidenced a significant correlation between BMI and concentrations of creatinine, urea, cystatin C, GFR estimated by the Filler formula, microalbuminuria, and the number of decayed teeth D. The strongest correlation was revealed between BMI and concentration of uric acid (r = 0.737), and oral variables such as PCR and BOP (r = 0.539; r = 0.583). Additionally, the most significant correlation was found between BMI Z-score and oral status estimated by BOP and DMFT scores (r = 0.499; r = 0.499). In the study group, statistical analysis evidenced a significant correlation between uric acid and HDL, triglycerides, and BOP. The strongest correlation was revealed between UA and LVMI (r = 0.616). There was a significant correlation between cystatin C and BOP, LVMI, and between BOP and triglycerides, glucose, HDL, and GFR estimated by the Filler formula (Table 3).

### 3.4. Clinical and Biochemical Characteristics: Suspicion of Obesity-Related Glomerulopathy Group and without ORG Group

Individuals composing the suspicion of ORG and without ORG group did not differ in serum concentrations of creatinine, urea, cystatin C and GFR estimated by the Filler formula. Significantly higher concentrations of uric acid were observed in the suspicion ORG group (*p* = 0.002). Mean GFR calculated by the Filler formula in the suspicion of ORG group was 130.14 ± 54.88 mL/min/1.73 m^2^ and 106.37 ± 20.38 mL/min/1.73 m^2^ in the without ORG group. Median microalbuminuria was significantly different between the probable ORG group and without ORG group (*p* = 0.001) (Table 4).

## 4. Discussion

### 4.1. Dental Caries

In this study, no specific relations between oral status and markers of kidney disease were found; however, more frequent gingivitis/bacterial colonization and significant differences in oral status between obese and non-obese children were revealed. Obese patients demonstrated a higher incidence of oral-related complications according to dental status and less efficient oral biofilm control and gingival inflammation than non-obese subjects.

Based on the literature review on childhood obesity, cause-and-effect relationships to oral diseases show strong associations with periodontal diseases and different relationships to caries progression, such as confirming [24,25,26] or denying [27,28,29,30]. It seems to depend on multiple factors contributing to the disease progress and the risk of dental caries in primary and permanent dentition among pediatric patients affected by simple obesity. This is, to the best of our knowledge, the third study related to obese children in our country [31,32]. In comparison to the first study carried out on obese children in the similar age group of 7 and 12 years old, in the group of younger children, no difference was found between healthy and obese children (82.2% versus 95.0%; ns), while in the group of children aged 12, the incidence of caries was significantly higher in the group of obese children (53.2% versus 84.2%; *p* = 0.004) [31]. In addition, obese adolescents revealed a correlation with the number of surfaces affected by caries, dental plaque, and gingivitis (*p* = 0.001). The second survey confirmed high caries incidence of obese children aged 6–12 years old, but the relationship between BMI and insufficient oral hygiene has not been demonstrated [32]. A similar study of 91 participants divided on normal and overweight children between 6–12 years old revealed a low caries experience and risk classification CAMBRA [30]. The lack of differences in the group consisted of younger children may be explained by the very high caries prevalence among Polish children under 7 years old, estimated at 85.0% [33].

In contrast, older children aged 12–18 have easier access to dental operatory offices and school oral health education/promotion facilities. Other co-factors such as preferred diet, socioeconomic class, industrialized origin life conditions may be found as moderators [29,34,35]. Assuming the present trial’s statistical differences between DMFT values at age 15, obese children revealed that increasing body weight boosts caries prevalence.

At this point, it is worth emphasizing the role of dentistry in the monitoring of childhood obesity [36,37]. Frequent contact with children and young adults in the dental setting makes it possible for them to prevent obesity. The overweight parent and child–dentist relationship in an extended period may allow that dietary advice may be given to reduce sugar consumption and meal frequency. This could be valid from the oral diseases and BMI reduction points of view. Moreover, becoming aware of the oral indicators of obesity may strengthen motivation to alter any unhealthy behaviors [36,37].

### 4.2. Dental Plaque and Gingival Bleeding

It is noteworthy that the relationship between obesity and periodontal inflammation is reported wider than dental caries and gingivitis experience, probably because of exposure to a number of factors [38,39,40]. Based on the literature mentioned above, a failure in periodontal health maintenance is due to an influence of blood proinflammatory cytokines, such as IL-1, IL-6, TNF-α, CRP, and oxidative stress activity [12]. Proinflammatory mediators secreted by adipose cells, interfering with the immune system, may modify the host’s response to plaque antigens and contribute to blocking periodontal protection [41]; therefore, focusing on gingivitis observed in obese children results from metabolic disturbances, inflammatory factors, and poor oral hygiene habits. Additionally, predisposition to gingivitis may be related to circulating sex hormones as higher estrogen levels owing to androgen disturbances in adipose tissue [42,43]. Even if a clinical survey did not show significant deviations in periodontal tissues, such as the number and depth of gingival pockets or periodontal attachment loss, poor cleaning habits with gingival inflammation may be observed among obese children [44].

At such a young age, periodontal health elements such as depth and attachment level are usually not examined because of their developmental status; however, it is known that high inflammatory scores and poor oral hygiene are more likely to generate permanent periodontal inflammation in the future. Moreover, the effectiveness of periodontal surgical treatment coexisting with obesity may have an uncertain recovery prognosis [45,46,47]. In this study, we demonstrated the adolescent patients affected by gingivitis, defined by the BOP index. This local manifestation of systemic inflammation accompanying obesity may suggest the development of metabolic syndrome and kidney damage in the form of obesity-related glomerulopathy.

Notably, obesity in pediatric patients may promote oral colonization with pathogens, which the PCR index may assess. This is in line with current literature concerning weight gain in childhood, where it is reported how excessive bacterial plaque increases unfavorable microbiota composition in the oral cavity [48,49]. In this context, Craig et al. revealed how oral microbiota trajectories varied in childhood according to weight gain [50]. Zeigler et al. documented an increase in both *Firmicutes* and *Bacteroidetes* sp. in obese patients compared to normal-weight individuals [43]. Nagawa et al. demonstrated that some species might support progesterone and estrogen access to vitamin K production needed for bacterial growth factors [51]. It seems that the role of oral bacteria in the formation and maintenance of inflammation in obesity may help to understand not only quantity but quality effects on microbial colonization or abundance in the oral cavity [50]. Changes leading to oral microbial dysbiosis need further microbiota investigations.

### 4.3. BMI and Obesity-Related Glomerulopathy

The study revealed significantly higher concentrations of uric acid, cystatin C, and mean GFR estimated by Filler formula in the obesity group than in the control group. These laboratory results showed the early, asymptomatic kidney injury without clinically important microalbuminuria in obese patients. The initial hypothesis could be that serum cystatin C and uric acid might aspire to be early markers of kidney damage in obesity, getting ahead of an increase in the concentration of creatinine, urea, and hypofiltration. Moreover, GFR calculated with the Filler formula based on serum concentration of cystatin C represents a more precise index of hyperfiltration, which is characteristic of ORG [52].

Additionally, our study presented the markers of metabolic syndrome in the study group and no hypertension, hypertriglyceridemia, low concentrations of HDL in the control group, which confirms the representative group of patients with obesity. In addition, uric acid is considered a marker connecting obesity with metabolic syndrome and is also indicative of the development of cardiovascular diseases, including arterial hypertension, atherosclerosis, and renal impairment [53]. The above reports are confirmed in the present study by a positive correlation between the uric acid level and markers of the metabolic syndrome: concentration of triglycerides, HDL, and LVMI indicating left ventricular hypertrophy and as a risk factor of cardiovascular diseases, including arterial hypertension.

In the present study, the BMI, describing simple obesity in children and adolescents, correlated the most with two oral status markers: PCR and BOP. Additionally, the project revealed a high correlation between BMI Z-score and gingivitis, estimated by the BOP index. The presented results may initiate the discussion and our future studies of the role of gingivitis in the progression of obesity. Moreover, the study evidenced a significant correlation between BOP index and concentration of kidney markers: uric acid, cystatin C, and indicators of metabolic syndrome: concentration of triglycerides, HDL, and glucose. Based on the above information and other reports [54], we suggest that gingivitis may represent an early marker of metabolic syndrome and kidney injury such as obesity-related glomerulopathy. In particular, the strongest correlation was found between the BOP index and uric acid. It seems to be the link between local inflammation in the oral cavity and systemic inflammatory response, which occurs in patients with obesity and morphological or functional changes in kidneys, described as obesity-related glomerulopathy.

Showing the usefulness of dental and biochemical markers, which are routinely assessed during diagnostics of obesity complications, may contribute to improved identification of the group of children with a risk of ORG and, therefore, chronic kidney disease in adulthood. In this way, invasive diagnostics of kidney damage, i.e., kidney biopsy, could be avoided. Early introduction of nephroprotective treatment and intensification of body weight reduction in this group of patients may significantly contribute to a decrease in the number of patients with obesity who may develop end-stage renal failure later in adulthood.

### 4.4. Limitations of the Study

The study has other limitations that have to be underlined in our interpretation of the results. No renal biopsies were performed in the group of patients with obesity-related glomerulopathy and healthy children and adolescents to confirm the clinical results received from the study. ORG is defined by glomerular hypertrophy (glomerulomegaly), adaptive focal segmental glomerulosclerosis, and tubulointerstitial fibrosis with tubular atrophy [55,56]. Renal biopsy is too invasive a procedure in children and adolescents (in Poland, it is usually used in case of acute renal failure without the onset point). At the time of the survey, the hormonal profiles of the subjects were not investigated, and oral status was examined without any pre-diagnostic dental history (baseline). Further, family members were not studied at all, and even they presented with overweight body mass. Finally, fewer than 100 participants may not be a representative study group for distinct conclusions. On the other hand, a homogenous group of children affected by obesity was collected, which was examined in one hospital and tested by the same diagnostic laboratory. Further, the assessment of salivary levels of such markers as cystatin C and uric acid should be considered in search of an alternative, still non-invasive method of ORG diagnosis in children [57]. The circumstances mentioned above would add valuable data to general analysis. Oral health indicators may motivate to begin obesity treatment and support diminishing the risk of developing or progressing obesity-related glomerulopathy and metabolic syndrome in pediatric patients. Obtained results highlight the differences in oral status between obese and non-obese young patients, as well as the need for prophylactic programs, especially in times of pandemic, with unpredictable access to public spaces and enforced lockdowns [58].

## 5. Conclusions

No specific relation to markers of kidney disease was found; however, more frequent gingivitis/bacterial colonization and significant differences in oral status between obese and non-obese patients were revealed. Susceptibility to inflammation may be conducive to developing metabolic syndrome and kidney damage in the form of obesity-related glomerulopathy and contribution to dental caries progression in the future.

Increased serum cystatin C and uric acid might aspire to be early markers of kidney damage leading to obesity-related glomerulopathy. GFR based on serum concentration of cystatin C represents a more precise index of hyperfiltration, which is characteristic of ORG. Uric acid seems to indicate metabolic syndrome and cardiovascular complications (LVMI > 95 percentiles).

Since the group with suspicion of ORG is small and comparisons with non-ORG patients are inconclusive, an analysis of the relationship between renal, metabolic, and oral health indicators among pediatric patients affected by simple obesity needs to be continued in future clinical trials.

## Figures and Tables

**Table 1 jcm-10-03723-t001:** Inclusion and exclusion criteria for study and control groups.

Criteria for Inclusion into the Obesity Group	Criteria for Inclusion into the Control Group	Criteria for Exclusion from Study and Control Groups
aged 9–18	aged 9–18	interview: prematurity, congenital abnormalities of urinary tract such as unilateral or bilateral kidney hypoplasia, unilateral kidney aplasia; incorrect kidney location in abdomen; unilateral, bilateral vesicoureteral refluxmedical history: recurrent urinary tract infections
simple obesity	normal weight	genetic obesity, diabetes mellitus, familial hyperlipidemia
normal blood pressure/primary hypertension	normal blood pressure	secondary hypertension (based on kidney diseases, coarctation of the aorta, endocrine disorders, iatrogenic–medications such as steroids)
BMI > 95 percentiles, BMI Z-score ≥ 1.4	BMI < 85 percentiles BMI Z-score < 1.3	BMI < 25 percentiles
		clinical or laboratory markers of previously acute or chronic diseases
		no aberrations in ECHO, USG
a patient, parent, or legal guardian approval	a patient, parent, or legal guardian approval	lack of acceptance from patients, parents, or legal guardians

Abbreviations: BMI—Body Mass Index, BMI Z-score—were transformed from BMI values using WHO reference values for pediatric BMI, ECHO—echocardiogram.

**Table 2 jcm-10-03723-t002:** Anthropometric, biochemical, and oral data for obesity group and control individuals.

Variables	Obesity Group (*n* = 45) Mean ± SD Median (Min-Max)	Control Group (*n* = 41) Mean ± SD Median (Min-Max)	*p*-Value
age [years]	15.40 ± 2.33	15.36 ± 2.74	ns
	16 (8–19)	16 (9–18)	
BMI [kg/m^2^]	34.05 ± 4.75	16.43 ± 2.43	0.001
	33.80 (26–47)	20.00 (15–24)	
BMI Z-score	2.30 ± 0.38	−0.10 ± 0.65	0.001
	2.32 (1.4–3.0)	0.07 (−1.7–0.74)	
creatinine [mg/dL]	0.63 ± 0.12	0.59 ± 0.19	ns
	0.63 (0.33–0.91)	0.57 (0.30–1.06)	
BUN [mg/dL]	24.17 ± 5.45	25.27 ± 6.42	ns
	24 (16–36)	24 (17–41)	
UA [mg/dL]	6.38 ± 1.33	4.40 ± 1.07	0.001
	6.70 (3.7–9.1)	3.90 (2.9–6.2)	
cystatin C [mg/L]	0.88 ± 0.16	0.76 ± 0.11	0.028
	0.91 (0.4–1.2)	0.75 (0.6–0.9)	
GFR F [ml/min/1.73 m^2^]	111.26 ± 31.15	127.18 ± 22.54	0.019
	102 (75–249)	127 (101–163)	
GFR S [ml/min/1.73 m^2^]	115.57 ± 20.54	118.02 ± 36.02	ns
	113 (80–175)	111 (53–199)	
microalbuminuria [mg/day]	20.56 ± 17.90	11.70 ± 7.03	ns
	14.17 (5–87)	10.92 (6–29)	
total cholesterol [mg/dL]	184.83 ± 33.67	173.08 ± 32.44	ns
	187 (112–260)	177 (122–243)	
HDL [mg/dL]	42.88 ± 7.90	55.47 ± 10.02	0.001
	42 (31–60)	56 (34–75)	
LDL [mg/dL]	112.79 ± 28.31	105.73 ± 27.55	ns
	113 (34–169)	103 (70–162)	
triglycerides [mg/dL]	161.28 ± 131.40	76.41 ± 25.39	0.001
	132 (44–785)	80 (27–130)	
glucose [mg/dL]	92.57 ± 6.14	88.79 ± 5.45	0.029
	91 (79–108)	89 (80–99)	
LVMI [g/m^2,7^]	37.30 ± 9.12	25.04 ± 3.28	0.001
	34.01 (29–54)	24.16 (20–29)	
D	1.91 ± 2.93	0.12 ± 0.33	0.001
	1 (0–11)	0 (0–1)	
M	0.08 ± 0.28	0 ± 0	0.054
	0 (0–1)	0 (0)	
F	2.03 ± 3.39	1.71 ± 1.94	ns
	1 (0–15)	1 (0–6)	
DMFT	4.03 ± 4.18	1.83 ± 1.99	0.016
	2 (0–16)	1.5 (0–7)	
d	0.40 ± 1.65	0.02 ± 0.15	ns
	0 (0–9)	0 (0–1)	
mt	0 ± 0	0.02 ± 0.15	ns
	0 (0)	0 (0–1)	
f	0 ± 0	0.26 ± 0.77	0.037
	0 (0)	0 (0–3)	
dmft	0.40 ± 1.65	0.31 ± 0.87	ns
	0 (0–9)	0 (0–3)	
DMFT + dmft	4.31 ± 4.39	2.14 ± 2.04	ns
	3 (0–17)	2 (0–7)	
PCR [%]	56.40 ± 34.16	15.87 ± 19.27	0.001
	50 (5–100)	9.5 (0–80)	
BOP [%]	46.46 ± 33.68	7.19 ± 10.64	0.001
	50 (0–100)	0 (0–40)	

Results are expressed as mean ± standard deviation, median (min-max ranges). Statistical significance is given according to *p*-value (*p* ≤ 0.05) vs. non-significance (ns). Statistical tests used were Mann–Whitney *U* test, *t*-test, and Welch test. Abbreviations: BMI—body max index; BUN—blood urea nitrogen; UA—uric acid; GFR F—glomerular filtration rate estimated by the Filler formula; GFR S—glomerular filtration rate calculated by the Schwartz formula; LVMI—left ventricular mass index, D—number of decayed secondary teeth; M—number of missing secondary teeth; F—number of filled secondary teeth; DMFT—decayed, missing, filled teeth score, evaluating dental caries in permanent teeth; d—number of decayed primary teeth; m—number of missing primary teeth; f—number of filled primary teeth; dmft score—total score evaluating the number of decayed, missing, filled primary teeth; PCR [%]—plaque control record index; BOP [%]—bleeding on probe index.

**Table 3 jcm-10-03723-t003:** Significant results of the Spearman’s and Pearson’s correlation rank tests regarding clinical and biochemical parameters (*p* < 0.05) for the obesity group.

Obesity Group *n* = 45	*p*-Value	Spearman R/Pearson r
BMI and D	0.001	R 0.433
BMI and PCR	0.001	R 0.539
BMI and BOP	0.001	R 0.583
BMI and creatinine	0.033	r 0.286
BMI and urea	0.007	R 0.256
BMI and UA	0.001	R 0.737
BMI and cystatin C	0.001	R 0.480
BMI and GFR F	0.001	R −0.524
BMI and microalbuminuria	0.004	R 0.419
BMI Z-score and D	0.025	R 0.316
BMI Z-score and BOP	0.001	R 0.499
BMI Z-score and DMFT	0.001	R 0.499
BMI Z-score and UA	0.000	r 0.606
BMI Z-score and cystatin C	0.000	R 0.477
BMI Z-score and GFR F	0.000	R −0.478
BMI Z-score and microalbuminuria	0.020	R 0.348
UA and BOP [%]	0.001	R 0.481
UA and HDL	0.001	r −0.582
UA and triglycerides	0.001	r 0.479
UA and LVMI	0.004	R 0.616
cystatin C and BOP [%]	0.005	R 0.407
cystatin C and LVMI	0.039	r 0.475
microalbuminuria and HDL	0.001	R −0.054
microalbuminuria and triglycerides	0.009	R 0.398
creatinine and HDL	0.014	R 0.337
BOP and GFR F	0.005	R −0.413
BOP and HDL	0.001	R −0.439
BOP and triglycerides	0.004	R −0.394
BOP and glucose	0.004	R 0.388
DMFT and total cholesterol	0.034	R 0.307
DMFT and triglycerides	0047	R 0.277
DMFT and glucose	0.041	R 0.279

Abbreviations: BMI—body max index; UA—uric acid; GFR F—glomerular filtration rate estimated by the Filler formula; LVMI—left ventricular mass index. D—number of decayed secondary teeth; DMFT—decayed, missing, filled teeth score, evaluating dental caries in permanent teeth; PCR [%]—plaque control record index; BOP [%]—bleeding on probe index.

**Table 4 jcm-10-03723-t004:** Anthropometric, biochemical, and oral data under suspicion of ORG (sORG) and without ORG among individuals.

Variables	sORG (*n* = 7) Mean ± SD Median (Min-Max)	Without ORG (*n* = 38) Mean ± SD Median (Min-Max)	*p*-Value
age	16.14 ± 1.77	15.16 ± 1.77	ns
	16 (14–19)	16 (8–18)	
BMI	35.78 ± 3.65	33.62 ± 4.94	ns
	36.6 (29.5–40)	32.61 (26–47)	
BMI Z-score	2.42 ± 0.25	2.28 ± 0.41	ns
	2.32 (2.06–2.7)	2.33 (1.36–3.03)	
creatinine	0.66 ± 0.09	0.62 ± 0.13	ns
	0.7 (0.53–0.77)	0.61 (0.33–0.91)	
BUN	23 ± 4.43	24.5 ± 5.71	ns
	22 (19–32)	24 (16–36)	
UA	7.31 ±0.59	6.14 ± 1.36	0.002
	7.3 (6.3–8.2)	6 (3.7–9.1)	
cystatin C	0.79 ± 0.21	0.9 ± 0.15	ns
	0.78 (0.41–1)	0.92 (067–1.2)	
GFR F	130.14 ± 54.88	106.37 ± 20.38	ns
	121 (92–249)	101 (75–144)	
GFR S	112.83 ± 19.19	116.26 ± 21.14	ns
	103 (92–144)	113.5 (80–175)	
microalbuminuria	51.41 ± 22.65	13. 95 ± 6.32	0.001
	41.5 (30–87)	13.45 (5–28)	
total cholesterol	195 ± 49.1	182.28 ± 29.3	ns
	218 (112–240)	185 (128–260)	
HDL	39.28 ± 6.68	43.78 ± 8.03	ns
	39 (33–51)	43 (31–60)	
LDL	122.28 ± 48.65	110.23 ± 20.64	ns
	141 (34–169)	111 (60–153)	
triglycerides	167 ± 56.07	159.85 ± 145.03	ns
	155 (108–259)	129 (44–785)	
glucose	91.43 ± 6.37	92.86 ± 6.17	ns
	95 (79–97)	91 (82–108)	
LVMI	53.4 ± 40	35.96 ± 8.08	ns
	53.4 (50–63)	33.06 (29–54)	
D	2. 14 ± 3.34	1.86 ± 2.89	ns
	0 (0–9)	1 (0–11)	
DMFT	3 ± 4.16	4.28 ± 4.21	ns
	1 (0–11)	2 (0–16)	
PCR [%]	67.42 ± 33.16	53.64 ± 34.44	ns
	80 (20–100)	50 (50–100)	
BOP [%]	50 ± 36.51	45.57 ± 33.59	ns
	50 (0–100)	45 (0–100)	

Results are expressed as mean ± standard deviation, median (min-max ranges). Statistical significance is given according to *p*-value (*p* ≤ 0.05) vs. non-significance (ns). Abbreviations: sORG—suspicion of obesity-related glomerulopathy. BMI—body max index; BUN—blood urea nitrogen; UA—uric acid; GFR F—glomerular filtration rate estimated by the Filler formula; GFR S—glomerular filtration rate calculated by the Schwartz formula; LVMI—left ventricular mass index. D—number of decayed secondary teeth; DMFT—decayed, missing, filled teeth score, evaluating dental caries in permanent teeth; PCR [%]—plaque control record index; BOP [%]—bleeding on probe index.

## Data Availability

Data associated with the paper are not publicly available but are available from the corresponding author at reasonable request.

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
