# Peer review of "Differences between Obese and Non-Obese Children and Adolescents Regarding Their Oral Status and Blood Markers of Kidney Diseases"

_jcm, 2021, doi:10.3390/jcm10163723_

Round 1

Reviewer 1 Report

The aim of the present study was to assess the relationship between renal, metabolic and oral health indicators among paediatric patients affected by simple obesity. Gingivitis, defined by the BOP index, and bacterial colonization, defined by the PCR index, were frequent observed among obese children. Susceptibility to inflammation may favour development of the metabolic syndrome, as well as kidney damage in the form of obesity-related glomerulopathy and contribution to a dental carious lesions in permanent teeth. Increased uric acid, cystatin C and microalbuminuria may play a predictive marker roles of kidney damage leading to obesity-related glomerulopathy. Uric acid seems to be an indicator of metabolic syndrome and cardiovascular complications (LVMI > 95 percentiles).

In my opinion, the conclusions are not evident from the analyses performed.  No specific assessment of the relation between oral status and markers of kidney disease have been demonstrated. No significant differences in BOP, PCR, DMFT and other indicators were found between ORG and no ORG groups. Moreover, a substantial language revision is required.

Author Response

Reviewer 1 comment: In my opinion, the conclusions are not evident from the analyses performed.  No specific assessment of the relation between oral status and markers of kidney disease have been demonstrated. No significant differences in BOP, PCR, DMFT and other indicators were found between ORG and no ORG groups.

Response by the Authors:

Thank you very much for the review of our manuscript and your positive assessment. We agree with you, our conclusions could be less hard. Therefore, we clarified conclusions in the Abstract (please see Page 1, lines 26-31) and Main File Conclusions (please see Page 12 Lines 401-412)

 Final Conclusions included to the Abstract: No specific relation between oral status and markers of kidney disease have been found, however gingivitis/bacterial colonization were frequent observed. Susceptibility to inflammation may be conducive to developing metabolic syndrome and kidney damage in the form of obesity-related glomerulopathy and contribution to dental caries progression. Uric acid seems to indicate metabolic syndrome and cardiovascular complications (LVMI > 95 percentiles).

Reviewer 1 comment: Moreover, a substantial language revision is required.

Response by the authors.

Thank you very much for your detecting of our language mistakes by translation process. Once again, we corrected English grammar errors and typos in the whole manuscript body.

Reviewer 2 Report

The work is original and the research is well detailed. However, revisions are needed. The authors should better explain the physio-pathological mechanism of renal damage in obese patients from microalbuminuria, a sign of renal hyperfiltration, at the genesis of chronic renal failure, also pointing out, as a limitation of the study, that in these patients no renal biopsy was performed.

Author Response

Dear Reviewer 2 

Thank you very much for the review of our manuscript and your positive assessment! Due to detailed explanations we have decided to add our response in attachment. as well in modified version of the manuscript file

Round 2

Reviewer 1 Report

As for my previous review, I don’t feel that the results are conclusive. No analysis in obesità related glomerulopathy have been performed. 

Author Response

Thank you very much for your email and the reviewers’ comments. We have carefully addressed all comments provided by the reviewers no 1. We feel that they have helped to improve our manuscript. In response to the comments, we made appropriate changes to the manuscript according to conclusions in the main manuscripts and abstract. The following modifications are included:

1/ the Title is changed concerning obese and non-obese children evaluation

2/ shortened Introduction

3/ paragraphs on histopathology in ORG is moved to Limitations of the study

4/ we have reconstructed conclusions in the Abstract and Main File and highlighted differences between obese and non-obese patients regarding their oral status. In the Discussion we have highlighted differences between obese and non-obese patients regarding their oral status, kidney effects. We have highlighted that only 3 surveys concerning obse and non-obese children in Poland were performed (Chlapowska et al 2014; Kaczmarek et al. 2014) and our study is the third one. We tried to compare to national study of children in the same age, but without subgroup of overweight group.

5/ four additional citations 1,2,5 and 58 are included (the rest of citations are in correct order)

All changes in the manuscript are highlighted in BLUE FONT color. Please find our point-by-point response in Review Section.

Yours sincerely, on behalf of all authors,

Katarzyna Mackowiak-Lewandowicz
